# Study on the formation mechanism of color and flavor during the processing of *Polygonatum cyrtonema Hua* (PCH)

Zhen Wang[1,2], Jin Xie[1,2], Mengshan Sun[1,2], An Liu[1], Li Zhou[1,2], Ye Yuan[1,2], Liu Cai[1,2], Rui Xu[1,2], Rong Song[1,2]*

**1** Hunan Institute of Nuclear Agriculture Sciences and Chinese Herbal Medicines, Hunan Academy of Agricultural Sciences, Changsha, Hunan, China, **2** Research Center for Medicinal Plants, Hunan Academy of Agricultural Sciences, Changsha, Hunan, China

* songrong0205@163.com

## Abstract

This study elucidated the formation mechanism of color and flavor in *Polygonatum cyrtonema Hua* (PCH) processing, corrected the traditional view that the Maillard reaction dominated, and verified the effects of processing on PCH's pharmacological activity and safety. Key parameters were monitored through colorimetric reactions and HPLC analysis during processing. Metabolic variations were analyzed by LC-MS, and volatile flavor compounds were examined via HS-SPME-GCMS. Antioxidant activity was assessed using DPPH and ABTS methods, and anti – aging efficacy was evaluated with a UVB-induced senescence model of HaCaT cells. The relationship between components and color/flavor profiles was explored through correlation analysis. The correlation coefficient between caramelization marker 5 – HMF and browning degree reached 0.96, much higher than that of Maillard marker CML (0.50). 5 – HMF accumulated significantly in the later processing stages (fourth and fifth drying cycles) up to $5.13 \pm 0.39$ mg/L, while CML increased only in the earlier phases. LC-MS analysis showed that fructose content significantly accumulated during processing, suggesting PCH's sweetness mainly comes from fructose. Fructose, a ketose, undergoes less enolization than aldose sugars like glucose, making it hard to bind with amino compounds for Maillard reaction. After the fourth and fifth drying cycles, amino acid content decreased by 30.70%, favoring caramelization over Maillard reaction in later stages. HS-SPME-GCMS revealed that PCH's primary aromatic compounds are oxygen – containing heterocyclics, which form during drying stages with content changes consistent with 5 – HMF accumulation. Maillard – related nitrogen – containing heterocyclic compounds have low content and minimal flavor contribution. Moreover, PCH antioxidant activity increased after processing, and its anti-aging effect was retained. In conclusion, contrary to traditional assumptions, the caramelization reaction in PCH processing is the primary process for color formation and contributes significantly to flavor development along with the Maillard reaction.

**Data availability statement:** All data files are available from the figshare database, the URL is https://doi.org/10.6084/m9.figshare.30039019. Any further data requests can also be sent via email to the staff at Hunan Institute of Nuclear Agricultural Sciences and Chinese Herbal Medicine (tangym1208@163.com).

**Funding:** This research was funded by the Agricultural Science and Technology Innovation Project of Hunan Province (2023CX63, 2024CX108), Earmarked fund for HARS-Chinese medicinal materials (HARS-11). The funders had no role in study design, data collection and analysis, decision to publish, or preparation of the manuscript.

**Competing interests:** The authors have declared that no competing interests exist.

This study analyzed the color and flavor formation mechanism and provided a theoretical basis for subsequent food development quality optimization.

## Introduction

*Polygonatum cyrtonema Hua* (PCH) belongs to the Polygonatum of the Asparagaceae and is widely consumed as a food and medicine in East Asia, notably in China [1,2]. PCH have been demonstrated to possess antioxidant [3], antidepressant [4], intestinal regulatory [5], anticancer [6], anti-aging [7], antihyperglycemic [8], and immunomodulatory [9] pharmacological activities as traditional foods or medicines. Based on industry statistics data, the annual output value of the PCH-related industry approaches $2 billion in China. The substances responsible for the primary pharmacological activities in PCH are water-soluble polysaccharides, which consist of mannose, galactose, rhamnose, arabinose, xylose, and glucuronic acid [10]. Galactose forms a linear main chain through 1,4-β glycosidic bonds, while rhamnose is embedded in the main chain via 1,2-α glycosidic bonds and forms a branch at the fourth position, primarily composed of arabinose, connected by 1,3-α or 1,5-α bonds [11]. The presence of glucuronic acid within the PCH polysaccharides significantly contributes to its antioxidant properties [12].

The primary edible components of PCH are the tubers. However, the fresh PCH tubers are not suitable for food or medicinal use, with the imperceptible sweet taste and flavor, obvious bitter taste, and weak pharmacological activity. Processed PCH exhibit a brownish-black color rather than their original yellow, carry a pleasant flavor and sweet taste, and have been proven to have improved antioxidant [13], intestinal regulatory [14], and antidepressant [15] effects. The traditional processing methods for PCH involve steaming and drying, which undergo repeated cycles to achieve the desired appearance and quality of the final products.

In practical production, the color and flavor of processed PCH are considered indicative of their quality. Public studies suggest that the color and flavor of PCH derive from Maillard reaction during the steaming process [16]. Monosaccharides and amino acids undergo that process of condensation, polymerization, and hydrolysis, resulting in the formation of advanced glycation end products (AGEs, such as carboxymethyl lysine), heterocyclic compounds (such as pyrazines), aldehydes, ketones, and melanoidins [17–19], which contribute to the color and flavor of processed PCH. However, this hypothesis has some limitations. Firstly, Maillard reaction and caramelization reactions could occur under the same conditions, but the substrates and products of the two reactions are different. The Maillard reaction uses carbonyl compounds and amino compound as substrate, and the product contains a large number of nitrogen-containing compounds, such as pyrazine [20], carboxymethyl lysine (CML) [21], and acrylamide [22]. The caramelization reaction can be based on monosaccharides such as fructose, and the products mainly include furfural (such as 5-Hydroxymethyl-2-furaldehyde, 5-HMF), maltol, aldehydes, and ketones [23]. These compounds can be produced through caramelization reaction in the absence of amino compounds [24].

Although the Schiff base obtained through the Amadori rearrangement in the Maillard reaction can also yield a slight amount of 5-HMF under alkaline conditions, it does not play a significant role in the products of the Maillard reaction. The two reactions have not been distinguished in the present studies, which may lead to inappropriate research results. Secondly, in the current hypothesis, water-soluble polysaccharides and their hydrolysates are considered primary substrates in the Maillard reaction, while the contribution of water-insoluble polysaccharides without pharmacological activity, such as cellulose and hemicellulose, is not considered. This implies that the color and flavor formation of processed PCH is surely based on the consumption of water-soluble polysaccharides [25], and the processing will inevitably lead to a significant reduction in the pharmacological activities of PCH [26]. Finally, AGEs and heterocyclic compounds produced by the Maillard reaction present adverse effects on the human body [27,28]. The color and flavor of the processed PCH, produced by the Maillard reaction, suggest that the processing of PCH is accompanied by food safety risks.

The inference of the current of PCH color and flavor formation mechanism is inconsistent with the actual situation that processed PCH are a safe food with better pharmacological activity. Therefore, the that the color and flavor of processed PCH are formed by the Maillard reaction is considered to be insufficient. To explain why the processed PCH obtained through steaming and drying are effective and safe, this study was conducted to elucidate and refine the mechanisms of color and flavor formation in processed PCH.

## Materials and methods

### Materials

The cultivated PCH were obtained as experimental plant materials from the Hunan Medicinal Plant Germplasm Resource Garden, Hunan Province, China. The PCH rhizomes were washed, grouped, and then processed alternately by steaming and drying. The grouping and processing methods for each group are described in S1 Table. Steaming conditions were set at 105°C for 1 h, and drying conditions were set at 55°C for 6 h.

DPPH free radical scavenging capacity assay kit, ABTS free radical scavenging capacity assay kit, and protein standard solution (5 mg/ml Bovine Serum Albumin) were purchased from Beijing Solarbio Science & Technology Co.,Ltd. (Beijing, China). 5-hydroxymethyl furfural (5-HMF), sulfuric acid, phosphoric acid, methanol, ethanol, anthrone, and Coomassie brilliant blue G-250 were purchased from Shanghai Macklin Biochemical Technology Co., Ltd. (Shanghai, China).CML ELISA kit was purchased from SenBeiJia Biological Technology Co., Ltd. (Nanjing, China). Total flavonoids quantitative kit and free amino acid quantitative kit were purchased from Suzhou Grace Biotechnology Co.,Ltd (Suzhou, China).

### Analysis of browning degree and lightness

The absorbance of the PCH water extract at 420 nm is utilized as a measure of the browning degree. Accurately weigh 1.0 g of PCH sample, add 10 mL of $H_2O$, and heat at 100 °C for 1h. Filter and transfer 200 µL of the supernatant into a 96-well plate for the detection of absorbance at 420 nm. Randomly selected freeze-dried, non-pulverized treated PCH samples were imaged for their external appearance using an image acquisition system. The acquired images were analyzed using Image-Pro Plus 6.0 (Media Cybernetics, Maryland, USA) to determine the lightness of each sample. The mean lightness values were calculated for each group, and the relative lightness of the G1-G10 groups was computed by using the lightness of the G0 group as the standard.

### Carbohydrate quantification

Accurately weigh 5.0 g of each sample and add 50.0 mL of 80% ethanol. Extract at 50°C for 2 h, filter, and repeat the process three times. Combine the filtrate and recover the ethanol under reduced pressure. Then, add double-distilled water (ddH$_2$O) to adjust the volume to 100.0 mL and extract at 100°C for 2 h. Filter the solution, concentrate the filtrate,

and lyophilize to obtain a powder for the determination of monosaccharide, oligosaccharide, and glycoside contents. For the residue from the ethanol extraction, add 100.0 mL of ddH$_2$O and extract at 100°C for 2 h, filter, concentrate the filtrate, and lyophilize to obtain a powder for the determination of water-insoluble polysaccharides. Dry the residue and pulverize it for the determination of water-insoluble polysaccharides. Carbohydrate quantification was performed using the sulfuric acid-anthrone method. The detection process is shown in Fig 1.

## Quantification of protein

Accurately weigh 1.0 g of the PCH sample, add 10 mL of H$_2$O, and perform ultrasonication for 15 min. Centrifuge the mixture at 12,000 rpm for 10 min, and filter the supernatant for testing. Accurately weigh 0.1 g of Coomassie Brilliant Blue G-250, dissolve in 95% ethanol, and add 10 mL of 85% (m/V) phosphoric acid. Make up the solution to 100 mL with ultrapure water to prepare the Coomassie Brilliant Blue working solution. Mix 1 mL of the PCH sample with 20 µL of the Coomassie Brilliant Blue G-250 working solution to quantify the protein content. Allow the mixture to stand for 5 minutes after thorough mixing, and then measure the absorbance at 595 nm. Using standard protein solutions, an identical reaction was conducted to quantify the protein content in each sample, and the relative protein content of the G1-G10 groups was computed by the G0 group as the standard.

## Analysis of antioxidant activity

The antioxidant activity of PCH simples was evaluated by DPPH and ABTS radical scavenging activity. PCH samples were extracted by 100°C water bath for 2 h, filtered, concentrated by rotary evaporation and freeze-dried. Accurately weigh 0.1 g of PCH extractive, add 2 mL of extract provided in kit, and heat at 40 °C for 30 min, the test solution with a concentration of 0.05 g/mL was obtained. Centrifugation at 10000 rpm for 10 min, and filter the supernatant for detection DPPH and ABTS radical scavenging activity. DPPH and ABTS radical scavenging activity was tested in strict accordance with the instructions. The detection wavelength of DPPH radical scavenging activity was set at 515 nm, and the detection wavelength of ABTS radical scavenging activity was set at 405 nm.

## Quantification of 5-HMF

Accurately weigh 1.0 g of the PCH sample, add 10 mL of methanol, and perform ultrasonic extraction for 1 h. Centrifugation at 12000 rpm for 3 min. The supernatant was then analyzed by SHIMADZU NEXERA X2 UPLC system after passing

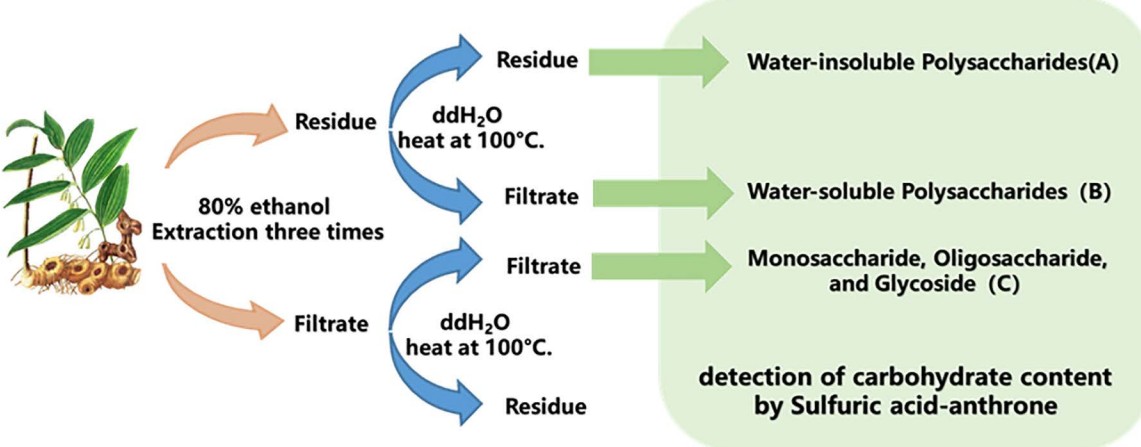

**Fig 1. Carbohydrate detection process.**

through a 0.22 μm microporous membrane. The chromatographic column employed was Agilent Extend C18 column (2.1×50 mm, 1.7 μm). The mobile phase consisted of methanol (A) and 0.1% phosphoric acid in water (B). The gradient program was as follows: starting at 90% B, decreasing to 85% B over the next 10 minutes, and holding for 10 minutes. The flow rate was set at 1 mL/min, and the injection volume was 10 μL. The detection wavelength was set at 284 nm. The 5-HMF content in processed PCH samples was quantified by external standard me·thod using 5-HMF reference material.

The polysaccharides of PCH were isolated and purified by water extraction and alcohol precipitation method combined with Sevag method. The treatment was repeated for 5 cycles under the identical processing conditions applied to the original PCH samples, with the exclusion of amino compounds such as free amino acids and proteins. After freeze-drying, 1.0 g of the obtained polysaccharide sample was dissolved for the determination of 5-HMF content. Subsequently, 7% of amino compound (lysine) – a ratio close to that of polysaccharides to free amino acids in the raw PCH sample – was added to the purified PCH polysaccharides. The same 5-cycle treatment was conducted, and the content of 5-HMF was determined thereafter.

## Quantification of free amino acids, total flavonoids and CML

Accurately weigh 1.0 g of PCH sample, add 10 mL of $H_2O$, and heat at 100 °C for 1h. Filter the supernatant for the detection of free amino acid content. The quantification of free amino acid was conducted with strict adherence to the protocols outlined in the kit's instruction manual. The reaction principle is based on the reaction of α-amino acids with ninhydrin hydrate. The detection wavelength was set at 570 nm, and the relative free amino acids content of the G1-G10 groups was computed by the G0 group as the standard. Accurately weigh 1.0g of PCH sample, add 10 mL of methanol, ultrasonic extract for 1h. Filter the supernatant for the detection of total flavonoids content. The quantification of total flavonoids was conducted with strict adherence to the protocols outlined in the kit's instruction manual. The reaction principle is based on the reaction of flavonoids with aluminum salt. The detection wavelength was set at 470 nm. Accurately weigh 1.0 g of PCH sample, add 10 mL of extract (N-butanol: methanol: $H_2O$ = 5:3:2), ultrasonic extract for 1 h. Filter the supernatant for the detection of CML content. The quantification of CML was conducted with strict adherence to the protocols outlined in the kit's instruction manual.

## Analysis of differential metabolites by LC-MS

Accurately weigh 1.0 g of each sample and add 10.0 mL of 70% methanol. Perform ultrasonic extraction, followed by centrifugation at 12,000 rpm for 3 min. The supernatant was then analyzed by LC-MS after passing through a 0.22 μm microporous membrane. All samples were separated using UPLC with a Waters Acquity Premier HSS T3 column (1.8 μm, 2.1 mm × 100 mm). The mobile phase consisted of 0.1% formic acid in water (A) and 0.1% formic acid in acetonitrile (B). The gradient program was as follows: starting at 5% B for 2 min, increasing to 60% B over the next 3 min, then to 99% B in 1 min and holding for 1.5 min, followed by a return to 5% B within 0.1 min and holding for 2.4 min. The flow rate was set at 0.4 mL/min, and the injection volume was 4 μL. MS analysis was performed using both positive and negative ion modes. Data acquisition was operated in the information-dependent acquisition (IDA) mode using Analyst TF 1.7.1 Software (Sciex, Concord, ON, Canada). The source parameters were set as follows: ion source gas 1 (GAS1), 50 psi; ion source gas 2 (GAS2), 50 psi; curtain gas (CUR), 25 psi; temperature (TEM), 550 °C; declustering potential (DP), 60 V or −60 V in positive or negative modes, respectively; and ion spray voltage floating (ISVF), 5000 V or −4000 V in positive or negative modes, respectively. The TOF MS scan parameters were set as follows: mass range, 50–1000 Da; accumulation time, 200 ms; and dynamic background subtract, on. The product ion scan parameters were set as follows: mass range, 25–1000 Da; accumulation time, 40 ms; collision energy, 30 V or −30 V in positive or negative modes, respectively; collision energy spread, 15; resolution, UNIT; charge state, 1–1; intensity, 100 cps; exclude isotopes within 4 Da; mass tolerance, 50 ppm; and maximum number of candidate ions to monitor per cycle, 18.

 

## Analysis of differential metabolites by HS-SPME-GCMS

Accurately 1.0 g of PCH samples were weighed and ground into a powder in liquid nitrogen. Subsequently, 0.5 g of the powder was immediately transferred to a 20 mL head-space vial (Agilent, Palo Alto, CA, USA) containing a saturated NaCl solution to inhibit any enzyme reactions. The vial was sealed, and SPME analysis was conducted. Each sample was incubated at 60°C for 5 min, followed by exposure of a 120 μm DVB/CWR/PDMS fiber (Agilent) to the headspace of the sample for 15 min at 60°C. Afterward, the samples were analyzed by GC-MS. Desorption of the VOCs from the fiber coating was performed in the injection port of the GC apparatus (Model 8890; Agilent) at 250 °C for 5 min in splitless mode. The identification and quantification of VOCs were carried out using an Agilent Model 8890 GC and a 7000D mass spectrometer (Agilent), equipped with a 30 m × 0.25 mm × 0.25 μm DB-5MS (5% phenyl-polymethylsiloxane) capillary column. Helium was used as the carrier gas at a linear velocity of 1.2 mL/min. The injector temperature was maintained at 250 °C. The oven temperature was programmed to start at 40°C (held for 3.5 min), then increased at 10°C/min to 100 °C, at 7 °C/min to 180 °C, and at 25 °C/min to 280 °C, where it was held for 5 min. Mass spectra were recorded in electron impact (EI) ionization mode at 70 eV. The quadrupole mass detector, ion source, and transfer line temperatures were set at 150, 230, and 280 °C, respectively. The mass spectrometer was operated in selected ion monitoring (SIM) mode for the identification and quantification of analytes.

## Analysis of anti-aging effects on PCH samples by SA-β-Gal

HaCaT cells in logarithmic growth phase were inoculated at $5 \times 10^4$ cells per well into 6-well plates and cultured for 24 h in a 37°C, 5% $CO_2$ incubator. After discarding the culture medium, the cells were washed twice with PBS. Except for the blank group, all other groups received 20 mJ/cm² UVB irradiation to establish an aging model. Following irradiation, the blank and model groups received DMEM medium containing 10% fetal bovine serum, while the experimental group received the same medium supplemented with 100 μg/mL PCH Sample. The cultures were continued for 48 h. After discarding the supernatant, the cells were washed twice with PBS, stained with SA-β-galactosidase solution, and incubated at 37°C in the dark for 12 h. Microscopic images were captured after staining, and Image Pro Plus 6.0 was used to analyze the images, calculating the staining intensity and coloration rate of different groups to evaluate the antagonistic effect of PCH.

## Statistical analysis

The statistical analysis of the results was conducted using SPSS version 25.0 (SPSS, Inc., Chicago, IL, USA), and the least-significant difference method was used. The data was visualized using GraphPad Prism 8 (GraphPad, Inc., San Diego, CA, USA) and Origin 9.0 (OriginLab Corporation, Northampton, MA, USA). Image analysis was used Image Pro Plus 6.0 (Media Cybernetics, Inc., Sarasota, Florida, USA). All data, including LC-MS, HS-SPME-GCMS, 5-HMF, CML, and antioxidant assays, were obtained through three independent replicate experiments. The original data obtained by mass spectrometry were calculated Variable Importance in Projection (VIP) by OPLS-DA model, and the Fold Change (FC) from univariate analysis was further combined to screen for differential metabolites. The criteria for differential metabolites were $\log_2 FC > |1|$ and VIP > 1.

## Results and analysis

### Browning degree and brightness change of processed PCH

The appearance, browning degree, and brightness of the processed PCH samples are depicted in Fig 2. The volume of the processed PCH samples significantly decreased, accompanied by a significant color change from pale yellow to brownish-black (Fig 2A). This is similar to the results of zhang et al. [25]. The relative lightness of the PCH samples decreased from 100% in G0 to 28.35% in G10. There was a significant reduction in relative lightness after each drying

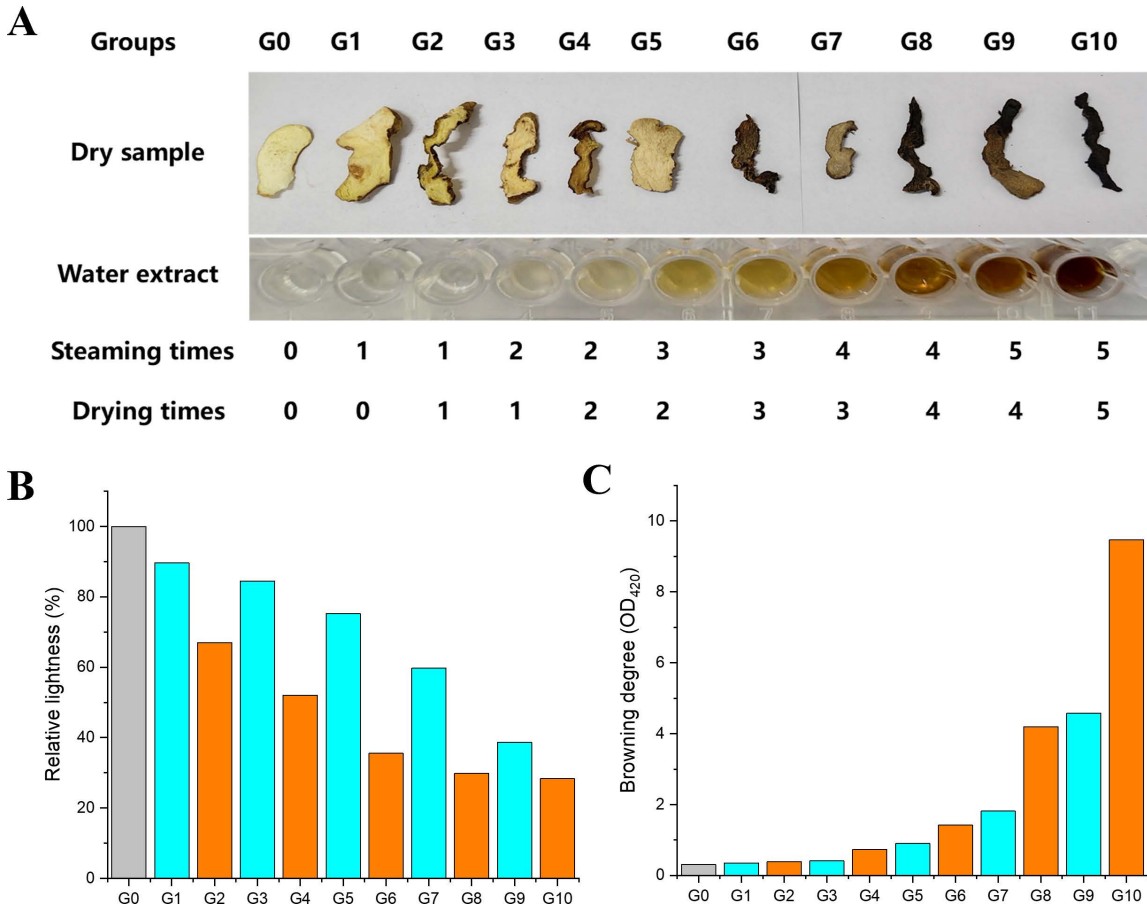

**Fig 2. Appearance, browning degree, and brightness change of processed PCH.** **(A)** The appearance of each processed PCH groups. **(B)** The relative lightness of each processed PCH groups. **(C)** The browing degree of each processed PCH groups. Gray indicates no-processed PCH, blue indicates the last processing step of the group is steaming, and orange indicates the last processing step of the group is drying.

stage, with a slight recovery following steaming (Fig 2B). This may be related to the loss of 5-HMF in the surface of PCH samples during steaming, which contributes to the color [29]. The browning degree of the PCH samples significantly increased from 0.31 in G0 to 9.47 in G10, with the drying stages being the primary contributor, and the browning degree change caused by the fourth and fifth rounds of processing is significantly stronger than that caused by the first three rounds of processing. Unlike the relative lightness, there was no decrease in browning degree after steaming stages (Fig 2C). These results suggest that drying is the primary factor for the color change in processed PCH samples, and this alteration may be associated with the dehydration of certain substances. Concurrently, the accumulation of substances may serve as a prerequisite for the change of color.

## Carbohydrate, protein and free amino acid contents change of processed PCH

The carbohydrate, protein, and amino acid contents of the processed PCH samples are showed in S1 Fig. Both water-insoluble and water-soluble polysaccharides continuously decreased during processing, with reductions of 27.99% and 62.38%, respectively, while the total contents of monosaccharides, oligosaccharides, and glycosides increased by 697.53%, It conforms to the law of compounds change given in the published literature [30]. These results indicate

that the macromolecular carbohydrates were transformed into smaller molecular carbohydrates during processing, with water-soluble polysaccharides being the primary carbohydrates subject to hydrolysis. The total carbohydrate content decreased by 21.15%, indicating that in addition to hydrolysis, there was concurrent loss or transformation of carbohydrates, such as loss caused by steaming or conversion into non-carbohydrate substances by the Maillard reaction or caramelization reactions.

Similarly, the protein content continuously decreased throughout processing, with a total reduction of 32.26%, indicating that proteins were also degraded to produce amino acids. However, there was no corresponding increase in amino acids, which remained largely unchanged during the first three processing rounds and then decreased by a total of 30.70% in the fourth and fifth rounds. The trend in amino acid changes correlates with the browning degree, suggesting that amino acids may be one of the substrates involved in the color formation of processed PCH samples.

## Total flavonoids content and antioxidant activity of processed PCH

The total flavonoid content of the processed PCH samples is shown in S2 Fig. Processing has increased the total flavonoid content in the PCH samples from 0.06% to 0.34%. This is consistent with the flavonoid changes caused by PCH processing [30]. This increase in total flavonoid could be due to the degradation of water-insoluble polysaccharides, such as cellulose and hemicellulose, which constitute the tissue structure of PCH during processing, thereby facilitating the dissolution of flavonoid. The decrease in water-insoluble polysaccharide content supports this. Flavonoid is generally considered one of the primary compound contributing to the antioxidative activity. The DPPH and ABTS radical scavenging activities of the processed PCH samples were significantly increased. Processing have increased the DPPH radical scavenging activity of PCH from 21.31±1.56% to nearly 100%, and the ABTS radical scavenging activity from 38.15±1.63% to 73.73±3.68%. The trends for both correspond to the alterations in total flavonoid content, illustrating that flavonoids are indeed one of the primary compound contributing to the antioxidative activity of the processed PCH samples.

## Anti-aging effects of processed PCH

The anti-aging efficacy of *Polygonatum cyrtonema Hua* stands as one of its primary pharmacological properties [7]. The anti-aging efficacy of PCH samples was evaluated using the UVB-induced HaCaT aging model, with results is shown in S2 Table. Under experimental conditions, the model group (MOD) showed a 166.26%±3.03% increase in staining rate compared to the control group (CTR). However, the intervention group using PCH samples exhibited significantly lower staining rates which present staining rateis between 84.10% and 110.08%.. These results indicate that processing did not substantially reduce the pharmacological activity of Polygonatum, including its anti-aging and antioxidant properties.

## 5-HMF and CML contents of processed PCH

The 5-HMF and CML contents of the processed PCH samples are shown in S3 Fig. The results are consistent with the conclusions of existing public literature [25]. The content of 5-HMF in the reaction products of the isolated polysaccharides is shown in S4 Table. 5-HMF is considered a marker product of the caramelization reaction, produced by the dehydration of fructose or glucose. Processing increased the 5-HMF content in the PCH samples from below the detection limit to 5.13±0.39 mg/L, with the increase primarily originating from the fourth and fifth drying stages. The reaction of PCH polysaccharides (carbonyl compounds) yielded 5-HMF at a concentration of 10.80±0.31 mg/L. When lysine (amino compound) was added to the system, the 5-HMF content was increased to 12.14±0.45 mg/L. Both reaction systems produced a color similar to that observed after the processing of PCH samples. Additionally, CML is deemed an AGEs produced during food processing, arising from the reaction between glucose and lysine in the Maillard reaction. Processing increased the CML content in the PCH samples from 33.19±5.80 pg/mL to 820.94±13.03 pg/mL, and the increase was primarily attributed to the first three rounds of processing.

## Correlation analysis

The correlation coefficients between the appearance and compositional changes of PCH samples during processing are showed in Fig 3. The correlation coefficients between proteins, water-soluble polysaccharides, and water-insoluble polysaccharides are all above 0.85, indicating that the hydrolysis of proteins and carbohydrates occurs synchronously during PCH processing. The low correlation coefficient of 0.57 between 5-HMF, a marker for caramelization reactions, and CML, a marker for Maillard reaction, suggests that these two reactions do not occur concurrently. The browning degree of PCH samples is highly correlated with the content of 5-HMF (0.96), which is significantly higher than that of CML (0.50), indicating that caramelization reactions are the primary factors contributing to color formation in processed PCH, rather than Maillard reaction as generally reported in the literature. The correlation coefficients between 5-HMF with small molecular carbohydrates (0.85) and water-insoluble polysaccharides (−0.88) indicate that the monosaccharides derived from the hydrolysis of water-insoluble polysaccharides serve as substrates for caramelization reactions. Furthermore, the contribution of monosaccharides produced by the hydrolysis of water-insoluble polysaccharides, such as cellulose and hemicellulose, to the reaction is greater than that of water-soluble polysaccharides (0.78). Additionally, the correlation between browning degree and water-insoluble polysaccharides (−0.79) is stronger than that with water-soluble polysaccharides (−0.69), suggesting that the monosaccharides primarily responsible for the color formation in PCH are derived from water-insoluble polysaccharides. The correlation coefficients between CML with small molecular carbohydrates and amino acids are −0.85 and −0.92, respectively, both of which are substrates for Maillard reaction. This difference suggests that the formation process of CML, namely the Maillard reaction, may be rate-limited by the content of amino acids, causing the primary reaction to shift from the Maillard reaction to caramelization in the later stages of processing. The correlation

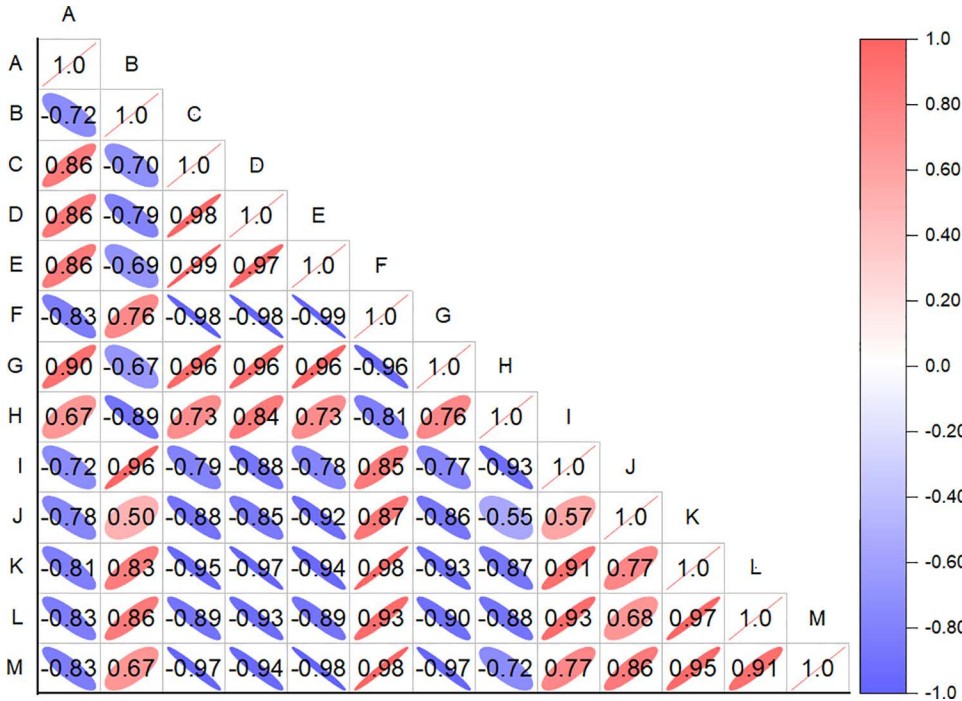

**Fig 3. Correlation of metabolite content changes of processed PCH. (A)** Brightness. **(B)** Browning degree. **(C)** Total carbohydrate. **(D)** Water-insoluble polysaccharides. **(E)** Water-soluble polysaccharides. **(F)** Monosaccharides, oligosaccharides, and glycosides. **(G)** Protein. **(H)** Amino acids. **(I)** 5-HMF. **(J)** CML. **(K)** Total flavonoids. **(L)** ABTS radical scavenging activity. **(M)** DPPH radical scavenging activity.

of heterocyclic compounds with caramelization and Maillard reaction was −0.29 and 0.01, respectively, suggesting that total heterocyclic compounds did not play a major role in the formation of color and flavor of PCH.

The monosaccharides and total flavonoid content are highly correlated with the DPPH and ABTS radical scavenging activities, both above 0.93. This implies that monosaccharides and total flavonoids provide antioxidant activity to processed PCH, with monosaccharides tending to contribute to DPPH radical scavenging activity (0.98) and total flavonoids to ABTS radical scavenging activity (0.97). The difference could stem from the distinct properties of the free radicals, unlike the neutral DPPH radical, the positively charged ABTS radical is more difficult to scavenged, thus the flavonoids are more inclined to scavenge ABTS radicals by the greater reducing power activity.

### LC-MS results of processed PCH

A total of 11 groups (from G0 to G10) were analyzed by LC-MS, resulting in the detection of 3149 metabolites. Significant differences were observed between the G0-G4 and G5-G10 groups, indicating that the initial two processing and the subsequent three rounds of processing exhibit significant differences (Fig 4A). The detected compounds include amino acids and derivatives (687), bnzene and substituted derivatives (433), organic acids (380), glycerides (185), alkaloids (176), alcohols and amines (158), heterocyclic compounds (139), flavonoids (108), phenolic acids (91), and other compounds (Fig 4B). PCA results revealed clearly distinct clusters among the different treatment groups, with steaming-baking treatment leading to significant and regular changes in the compositional profile of PCH. The PCH samples continuously increased along the PC1 and exhibited a trend of initial increase followed by a decrease along the PC2 (Fig 4C).

Setting the criteria to $\log_2 FC > |1|$ and VIP > 1 identifies differential metabolites. The number of differential metabolites in each group is depicted in Fig 4D. The results indicate that the initial two processing steps have a significant impact on the contents of PCH, with steaming exerting more significant influence than oven-drying. The differential metabolites common to G0-G10 are shown in Fig 4E, only one metabolite changed significantly in each step of processing, and there is no current evidence to suggest that this metabolite is associated with the color and flavor formation of processed PCH. The steaming and oven-drying processes, respectively, reveal three and five common differential metabolites (Fig 4F and 4G), and similarly, there is no evidence indicating that these substances may be related to the color and flavor formation of PCH.

The processing of PCH is characterized by deepening of the browning, softening of texture, increase in sweetness and aroma, and decrease in astringency. These modifications are possibly associated with alterations in the concentrations of carbohydrates, gallic acid, and tannins. The changes in the content of relevant metabolites after each processing steps is shown in Table 1.

The differences in carbohydrate content were negligible ($\log_2 TC < |0.27|$), with the exception of fructose, cellobiose, and xylobiose, which exhibited significant alterations with $\log_2 TC$ of 2.84, 4.70, and 1.35, respectively. It is inferred that the sweet taste in processed PCH is derived from fructose, which is produced by the degradation of fructan. Concurrently, cellulose and hemicellulose undergo substantial degradation during processing, yielding intermediates such as cellobiose and xylobiose, which are subsequently converted into monosaccharides that participate in Maillard and caramelization reactions, thereby softening the taste. Significantly reduced tannin ($\log_2 TC = −1.85$) during the processing of PCH may account for the alleviation of its astringent taste.

Processed PCH not only retains its sweetness but also acquires unique and agreeable flavors and aromas. Table 2 details the change of total content in the short peptides, aromatic compounds, alcohols and amines, carboxylic acids, organic acids, and heterocyclic compounds. Correlation analysis was also performed to assess the relationship between these substances and the indicators of Maillard or caramelization reaction. The analysis reveal a strong correlation between caramelization and alcohols/amines (0.87), and a significant correlation between the Maillard reaction and carboxylic acids and organic acids (0.80 and 0.86, respectively). The increased presence of these compounds is believed to be closely linked to the formation of the distinctive flavors and aromas in processed PCH.

 

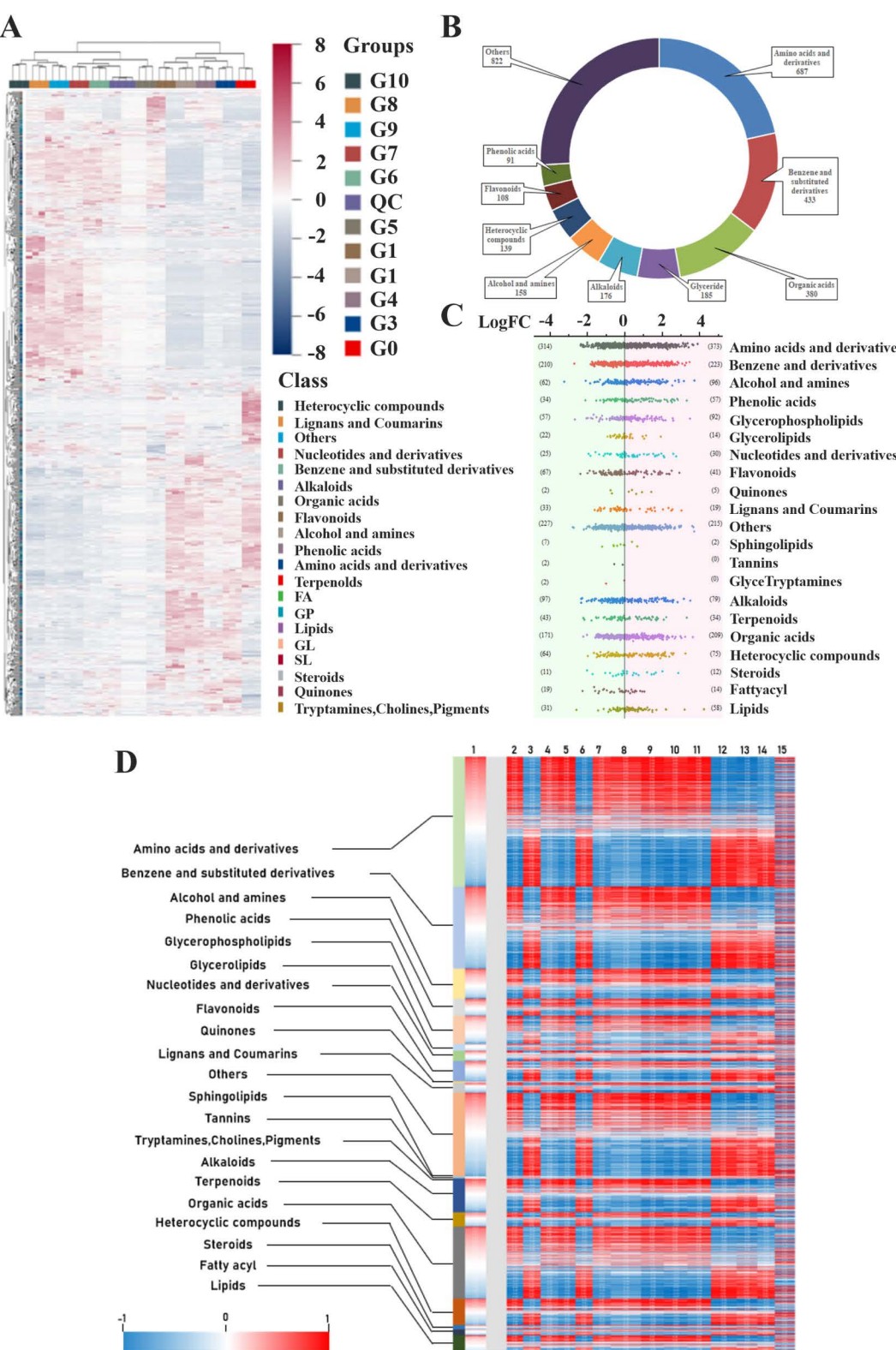

**Fig 4. Detected metabolites in processed PCH simples. (A)** Heat map of detected metabolites in processed PCH simples. **(B)** Classification and quantity of detected metabolites in processed PCH simples. **(C)** PCA analysis of LC-MS in processed PCH simples. **(D)** Differential metabolites caused

by each step of processing. **(E)** Venn diagram of differential metabolites caused by each step of processing. **(F)** Venn diagram of differential metabolites caused by steaming. **(G)** Venn diagram of differential metabolites caused by drying.

**Table 1. The content change trend of metabolites in PCH with processing.**

| Processing times | 1 | 2 | 3 | 4 | 5 | 6 | 7 | 8 | 9 | 10 | Log$_2$TC |
|---|---|---|---|---|---|---|---|---|---|---|---|
| Glucose | 0.49# | −0.07 | −0.14 | 0.23 | −0.56 | −0.59 | 0.39 | −0.08 | 0.03 | 0.03 | −0.27 |
| Mannose | −0.21 | 0.00 | −0.03 | 0.23 | −0.56 | 0.51 | −0.04 | 0.10 | 0.23 | −0.08 | 0.15 |
| Rhamnose | 0.13 | 0.03 | 0.04 | −0.06 | −0.17 | 0.14 | −0.10 | 0.01 | 0.02 | 0.02 | 0.06 |
| Fructose | 0.64 | −0.24 | 0.96 | 0.34 | 0.70 | 0.31 | −0.44 | 0.12 | 0.40 | 0.05 | 2.84 |
| Sucrose | −0.47 | −0.27 | 0.67 | 0.31 | 0.24 | −0.63 | 0.17 | −0.31 | 0.36 | −0.34 | −0.26 |
| Cellobiose | −0.70 | 0.26 | 2.56 | 0.65 | 0.89 | 0.84 | 0.00 | 0.33 | 0.34 | −0.47 | 4.70 |
| Xylobiose | 0.40 | 0.12 | 0.20 | −0.22 | 0.62 | 0.12 | −0.36 | 0.20 | 0.02 | 0.25 | 1.35 |
| 5-HMF | 0.20 | −0.73 | 4.35 | 0.40 | 1.07 | −0.07 | 0.98 | −0.39 | 0.69 | −0.09 | 6.42 |
| Tannin | 0.58 | −0.58 | 0.05 | −0.49 | −0.54 | 1.35 | −0.02 | −1.96 | 0.17 | −0.41 | −1.85 |
| Glucosamine | −2.46 | 0.78 | 1.44 | 0.98 | 3.24 | 0.25 | 1.91 | 0.36 | 0.36 | 0.57 | 2.70 |
| Disaccharide amine | 0.89 | −0.36 | 2.02 | 0.94 | 0.78 | 0.28 | −0.66 | −1.11 | 0.30 | −0.37 | 7.42 |

# The data in the table represent the difference in the content of the substance before and after each processing (Log$_2$FC), and Log$_2$TC represents the difference after the processing and before all processing.

**Table 2. Changes of metabolites related to aroma and taste.**

| Metabolites | G0 | G1 | G2 | G3 | G4 | G5 | G6 | G7 | G8 | G9 | G10 | Correlation coefficient | |
|---|---|---|---|---|---|---|---|---|---|---|---|---|---|
| | | | | | | | | | | | | VS 5-MFA | VS CML |
| 5-HMF (mg/L) | 0.00 | 0.00 | 0.01 | 0.14 | 0.23 | 0.81 | 0.83 | 2.06 | 2.12 | 3.45 | 5.13 | – | – |
| CML (ug/L) | 33.19 | 382.93 | 602.48 | 670.12 | 672.84 | 749.52 | 742.21 | 769.07 | 795.95 | 800.18 | 820.94 | – | – |
| short peptides# | 100.00% | 94.91% | 89.86% | 93.89% | 101.69% | 110.41% | 117.35% | 124.49% | 125.44% | 120.98% | 127.29% | 0.83 | 0.60 |
| Aromatic compound | 100.00% | 89.12% | 72.80% | 74.90% | 68.52% | 70.73% | 69.84% | 70.27% | 75.49% | 69.97% | 76.00% | −0.91 | −0.36 |
| Alcohols and amines | 100.00% | 69.79% | 67.60% | 62.82% | 110.42% | 131.41% | 115.35% | 120.63% | 124.60% | 137.11% | 128.54% | 0.87 | −0.36 |
| Carboxylic acid | 100.00% | 104.22% | 103.88% | 118.71% | 162.87% | 195.86% | 190.61% | 208.67% | 240.41% | 195.06% | 219.55% | −0.57 | 0.86 |
| Organic acids | 100.00% | 114.70% | 109.00% | 109.21% | 111.93% | 113.29% | 118.31% | 123.30% | 119.85% | 116.39% | 120.15% | 0.52 | 0.80 |
| Heterocyclic compound | 100.00% | 125.62% | 110.75% | 108.00% | 120.66% | 106.80% | 100.72% | 107.21% | 113.73% | 96.00% | 100.49% | −0.29 | 0.01 |

#The data in the table represent the relative content of the substance in each group of samples.

## HS-SPME-GCMS results of processed PCH

A total of 11 groups (from G0 to G10) were analyzed by HS-SPME-GCMS, resulting in the detection of 548 metabolites (Fig 5A), including terpenoids (104), esters (87), ketones (70), alcohols (69), heterocyclic compounds (58), aldehydes

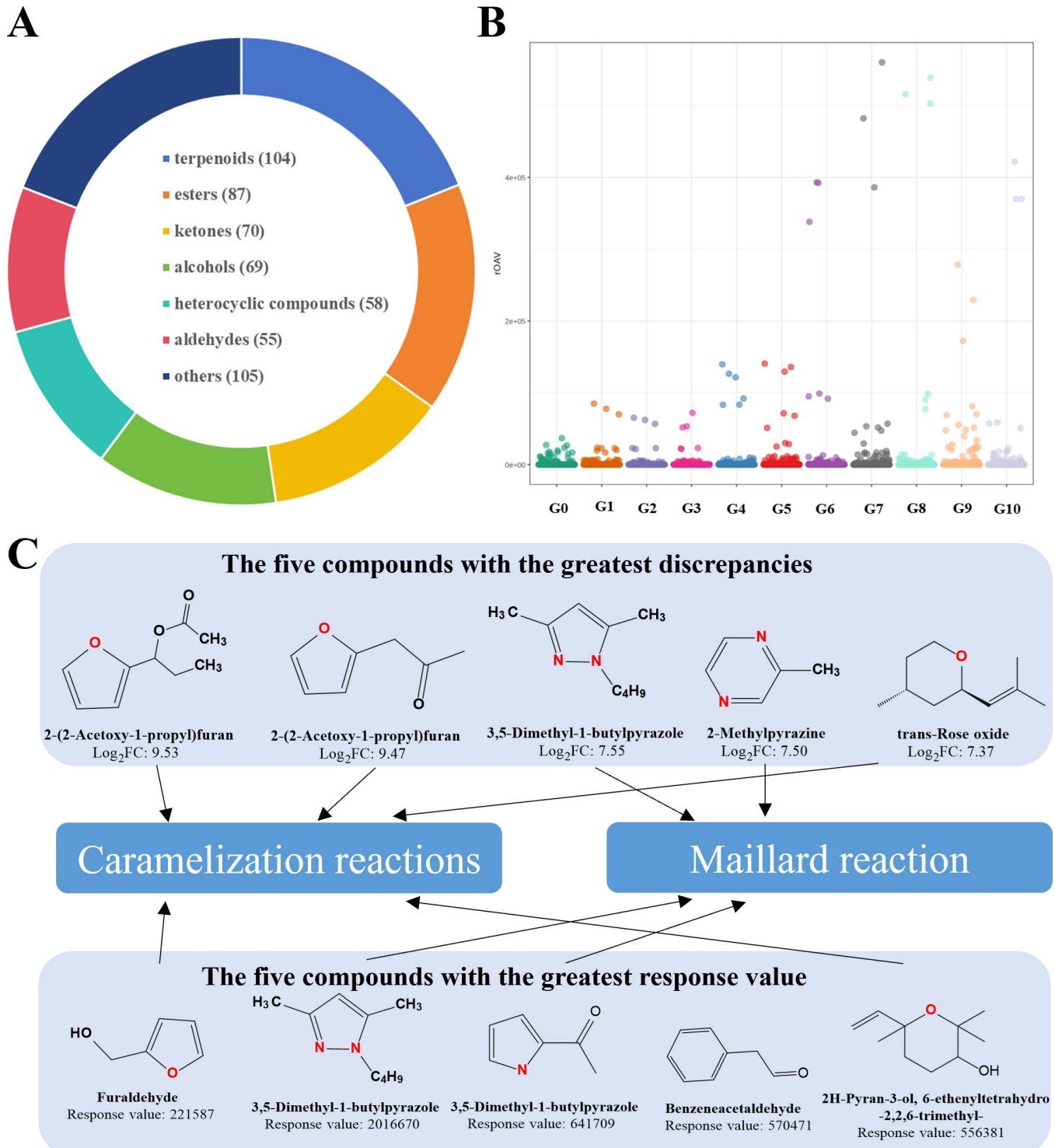

**Fig 5. HS-SPME-GCMS results of processed PCH. (A)** Classification and quantity of detected metabolites in processed PCH simples. **(B)** rOAV analysis of processed PCH simples. **(C)** Main volatile differential metabolites.

(55), and other compounds (105). By analyzing the aroma composition of processed PCH through the relative odor activity value (rOAV), it was found that the three carbohydrate breakdown products trimethylcyclohexene methyl mercaptan, ethyl cyclohexene formate, and nonadienal contribute the major aroma, and are mainly produced by the fourth and fifth rounds of processing (Fig 5B).

Setting the criteria to $\log_2 FC > |1|$ and VIP > 1 identifies differential metabolites, a total of 312 differential metabolites were obtained. The top five metabolites with the highest abundance and the five with the most significant compositional changes during processing are depicted in Fig 5C. Oxygen-containing heterocyclic compounds, such as furaldehyde, may be produced by caramelization reactions and are accompanied by caramel and fruity aromas. Nitrogen-containing heterocyclic compounds, such as 2-methylpyrazine, may be produced by Maillard reaction and are accompanied by ether like aroma. Considering the content changes, relative abundance, and aromatic characteristics, caramelization reactions are considered the primary process contributing to the formation of aroma in processed PCH, while Maillard reaction are secondary. It is noteworthy that the primary production of aromatic components from caramelization reactions occurs during the steaming stage, in contrast to the color development, which primarily takes place during the drying stage.

### Analysis of changes in content of representative compounds

Although the maillard reaction and the caramelization reaction occur at the same time, there are still analytically distinguishable products between them. We conducted further analysis of raw data from LC-MS and HS-SPME-GCMS, identifying multiple representative compounds to distinguish the processes of Maillard reaction and caramelization. Pyrazine derivatives and acrylamide serve as representative compounds for Maillard reaction, with the former produced through the condensation of α-diketo compounds with α-amino ketones [31], and the latter generated via dehydration of Amadori compounds [32]. Furfural and furan derivatives act as markers for caramelization reactions, both produced through processes like dehydration, cyclization, and cleavage of carbohydrates [25]. Changes in these compounds after PCH processing are shown in S3 Table. Results indicate that among compounds representing Maillard reaction, all except the extremely low-abundance dimethylpyrazine showed fold change ranges of 0.60–17.41. In contrast, compounds representing caramelization reactions, furfural and furan derivatives, displayed fold change ranges of 7.00–740.56, suggesting that caramelization dominates rather than Maillard reaction during PCH processing. This may be attributed to PCH polysaccharides being primarily composed of ketones like fructose, which are unfavorable for Maillard reaction initiation. In addition, it is worth noting that Maillard products such as acrylamide have certain safety risks [32]. According to the WHO technical report (Evaluation of certain food additives and contaminants), furfural and furan do not pose a direct safety risk.

## Discussion

### The formation mechanism of color and flavor during the processing of PCH

Traditionally, the color and flavor of processed PCH have been attributed to the Maillard reaction, which is supposed to occur during the steaming process. This is due to the higher temperatures employed during the steaming stage of PCH processing (approximately 100°C), which are within the range required for the Maillard reaction (80–140°C) [33], but lower than those necessary for caramelization reaction (150–170°C) [34]. However, it has been observed that caramelization reaction can occur even at lower temperatures [35], suggesting that color and flavor formation during the drying stage of processing could be attributed to caramelization. In this study, the PCH polysaccharide without amino compounds yielded 10.80±0.31 mg/L of 5 – HMF after processing, while the control group with added lysine produced 12.14±0.45 mg/L. Both reaction systems exhibited a color similar to that observed after the processing of the PCH sample. This demonstrates that the carbohydrates in PCH can undergo caramelization to produce 5 – HMF under low – temperature conditions without amino compounds [36]. Furthermore, it indicates that caramelization rather than the Maillard reaction is the primary cause of PCH coloration. The results indicate that the color, flavor, and sweet taste of processed PCH are

primarily contributed by the Maillard reaction, caramelization, and accumulated fructose, respectively (Fig 6). During the processing of PCH which is classic Similar to processing situations, carbohydrates and proteins are hydrolyzed during steaming stage, providing substrates for both the Maillard and caramelization reaction [37]. These reactions often occur concurrently during food processing and require monosaccharides as substrates. Additionally, the Maillard reaction requires amino acids, and the content of amino acids in PCH shows a correlation with the progression of the Maillard reaction. This suggests that in the early stages of processing, PCH contain a higher proportion of amino acids relative to monosaccharides, leading the Maillard reaction to dominate and produce nitrogen-containing heterocyclic flavor compounds such as pyrazines and pyrroles, which impart a unique aroma to the processed PCH. In the later stages of processing, as the concentration of monosaccharides increases and that of amino acids decreases, the caramelization reaction becomes predominant during drying, resulting in the production of flavor and color oxygen-containing heterocyclic compounds such as 5-HMF and furanones, which contribute to the flavor and color of the processed PCH. Furthermore, the accumulation of fructose during the the fourth and fifth rounds processing also facilitates a shift in the reaction from Maillard-dominated to caramelization-dominated.

In addition to sweet taste, processed PCH also produce a unique and pleasant aroma. The study reveals that the volatile substances crucial to the aroma of PCH predominantly consist of oxygen-containing heterocyclic and nitrogen-containing heterocyclic compounds. The nitrogen-containing heterocyclic compounds originate from the strecker degradation of α-amino compounds with α-dicarbonyl compounds. Following oxidative decarboxylation, α-amino compounds condense with α-dicarbonyl compounds to form pyrazines and other nitrogen-containing heterocycles, a process characteristic of the intermediate stages of the Maillard reaction. Conversely, oxygen-containing heterocyclic compounds are

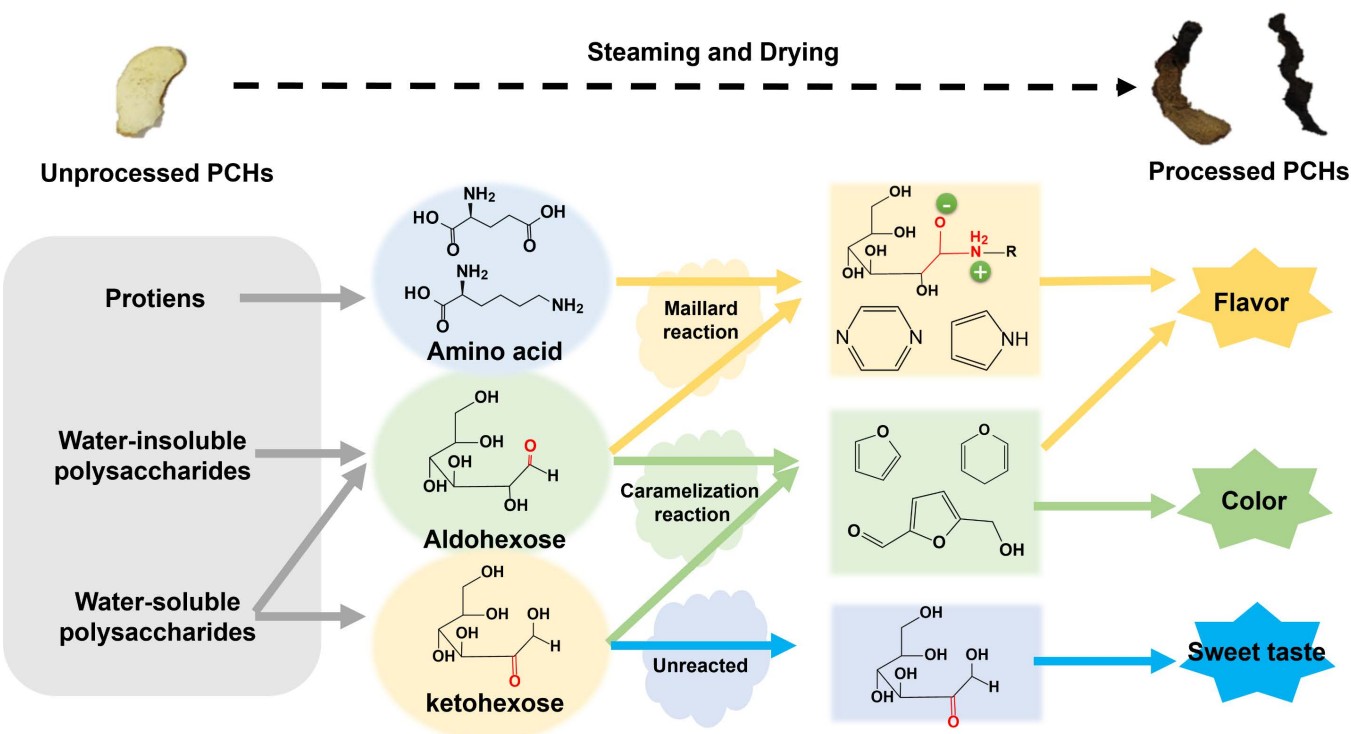

**Fig 6. Formation mechanism of color and flavor in PCH processing.**

 

formed through the condensation of α-dicarbonyl compounds and their degradation products, a process that should be attributed to caramelization reactions and is not inherently related to the Maillard reaction.

## The PCH color and flavor formation do not necessitate excessive loss of water-soluble polysaccharides with pharmacological activity

In the traditional processing paradigm of PCH, the formation of color is attributed to the participation of polysaccharides in the Maillard reaction following hydrolysis. Polysaccharides in PCH are pivotal constituents that provide pharmacological activity. This implies that the process of color and flavor formation in PCH is inextricably linked to the massive consumption of pharmacological activity. However, the actual outcomes do not corroborate this. This study revealed that the degradation products of different macromolecular carbohydrates could contribute differently to color formation. The macromolecular carbohydrates in PCH include water-insoluble carbohydrates such as cellulose and hemicellulose [38], as well as water-soluble carbohydrates such as fructans and polygalactose [39]. The hydrolysis products of the former are predominantly composed of aldohexose such as glucose, whereas the latter additionally include ketohexose such as fructose. The correlation analysis observed that water-insoluble polysaccharides contribute more significantly to color formation than their water-soluble polysaccharides. LC-MS results also indicate that, among the small molecular carbohydrates potentially involved in the Maillard reaction, only the fructose content increases significantly, while the other components exhibit minor variation. This discrepancy could attributed to the difficulty of monosaccharside participating in Maillard reaction. Glucose readily engages in the Maillard reaction as an aldohexose, whereas fructose, being a ketohexose, does not preferentially participate due to its lower degree of enolization compared to glucose in the system [40]. The glucose which derived from water-insoluble polysaccharides such as glucan, cellulose, and hemicellulose with weak pharmacological activity [41], participates in the Maillard and caramelization reaction, contributing to the color and flavor of PCH. And the fructose which derived from fructan, participates less in the Maillard reaction, thereby accumulating in the processed PCH and providing sweet taste. This indicates that the formation of color, flavor, and sweet taste in PCH does not necessarily require the hydrolysis of polysaccharides with pharmacological activity to provide the necessary substrates.

## The color is mainly formed by caramelization reaction suggesting that the processed PCH is not directly related to the accumulation of adverse substances

It is noteworthy that the color and flavor of PCH are primarily attributed to the Maillard reaction in traditional hypotheses, with adverse substances such as AGEs [42] and heterocyclic compounds [43] being responsible for these attributes. This suggests that the color formation of PCH necessarily produces adverse substances. However, the results of this study reveal that the sweet taste and color of processed PCH are not mainly derived from the Maillard reaction. The former primarily stems from the accumulated fructose by the hydrolysis of carbohydrates, and the latter primarily stems from dehydrated products such as 5-HMF by the caramelization reaction. 5-HMF has been reported to possess anti-inflammatory [44], immune-activating [45], antioxidant, and other pharmacological activities, which are beneficial for improving the healthcare effects of processed PCH. Simultaneously, its intake of adverse reactions is at a dose of 80–100 mg/kg [46]. After conversion, the maximum content of 5-HMF in processed PCH was approximately 5.13 mg per 100 g in this study. The recommended daily dosage of PCH is 10 g when it is administered as a traditional medicine, and the daily intake is generally no more than 500 g when it is consumed as a food. Consequently, the intake of 5-HMF through PCH consumption was no more than 0.026 g. Based on the estimation for a human body weight (about 60 kg), the intake of 5-HMF derived from PCH was far lower than the dose that can induce adverse reactions (approximately 4.8–6.0 g). It can therefore be concluded that 5-HMF in PCH does not pose any potential safety risks. Based on this, the study suggests that improving the rate of carbohydrate hydrolysis and caramelization reactions in the early stages of processing, while limiting the progression of the Maillard reaction, could obtain the desired color and flavor while reducing the accumulation of adverse substances, thereby improving the quality and safety of processed PCH as food and health products.

In this study, the mechanism by which PCH generate their characteristic colors and flavors during processing are attempted to explain, the substances that influence these attributes are analyzed. Nonetheless, the current research and analysis have yet to clearly elucidate the entire process of these substances from the initial substrate to the final product, due to the insufficiency of available data for such an analysis. We anticipate conducting related research in the future.

## Conclusion

This study clarifies the formation mechanism of color and flavor during the processing of PCH, correcting the traditional view that the Maillard reaction dominates. Caramelization is identified as the primary process for color formation, while it also contributes significantly to flavor development alongside the Maillard reaction. The sweet taste of processed PCH is mainly derived from accumulated fructose, and its flavor is predominantly provided by oxygen-containing heterocyclic compounds. Notably, PCH processing does not compromise its pharmacological activities; instead, it enhances antioxidant capacity and retains anti-aging efficacy. Additionally, the primary product of caramelization, 5-HMF, poses no safety risks within normal consumption doses, avoiding potential hazards associated with Maillard reaction-derived adverse substances. This research provides a theoretical foundation for the quality optimization and safety control of PCH-related food and health products.

## Supporting information

**S1 Table. The grouping of PCH samples.**
(PDF)

**S1 Fig. Carbohydrate, protein and free amino acid of processed PCH.** (A) The water-insoluble polysaccharides content of each processed PCH groups. (B) The water-soluble polysaccharides of each processed PCH groups. (C) The total monosaccharides, glycosides and oligosaccharides content of each processed PCH groups. (D) The total carbohydrate content of each processed PCH groups. E The relative protein content of each processed PCH groups. F The relative amino acid content of each processed PCH groups. (Gray indicates no-processed PCH, blue indicates the last processing step of the group is steaming, and orange indicates the last processing step of the group is drying. Different letters indicate significant differences in results at the $p < 0.05$.).
(PDF)

**S2 Fig. Total flavonoids content and antioxidant activity of processed PCH.** (A) Total flavonoids content of each processed PCH groups. (B) The DPPH and ABST radical scavenging rate of each processed PCH groups. (Gray indicates no-processed PCH, blue indicates the last processing step of the group is steaming, and orange indicates the last processing step of the group is drying. Different letters indicate significant differences in results at the $p < 0.05$.).
(PDF)

**S2 Table. The anti-aging efficacy of PCH Samples.**
(PDF)

**S3 Fig. 5-HMF and CML content of processed PCH.** (A) 5-HMF content of each processed PCH groups. (B) CML content of each processed PCH groups. (Gray indicates no-processed PCH, blue indicates the last processing step of the group is steaming, and orange indicates the last processing step of the group is drying. Different letters indicate significant differences in results at the $p < 0.05$.).
(PDF)

**S3 Table. Representative compounds detected by LC-MS and HS-SPME-GCMS.**
(PDF)

**S4 Table. 5-HMF content under the condition of PCH Polysaccharide in dependently processing.**
(PDF)

## Author contributions

**Conceptualization:** Zhen Wang, Rong Song.

**Data curation:** Zhen Wang, Ye Yuan, Rui Xu.

**Funding acquisition:** Rong Song.

**Methodology:** Zhen Wang.

**Supervision:** Rong Song.

**Validation:** Zhen Wang, Jin Xie, Mengshan Sun.

**Writing – original draft:** Zhen Wang.

**Writing – review & editing:** Zhen Wang, Jin Xie, An Liu, Li Zhou, Liu Cai.

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
