## [Decision Letter · Decision Letter 0]

26 Jul 2025

Dear Dr. Song,

Thank you for submitting your manuscript to PLOS ONE. After careful consideration, we feel that it has merit but does not fully meet PLOS ONE’s publication criteria as it currently stands. Therefore, we invite you to submit a revised version of the manuscript that addresses the points raised during the review process.

We look forward to receiving your revised manuscript.

Kind regards,

José M. Alvarez-Suarez

Academic Editor

PLOS ONE

Journal Requirements:

This research was funded by the Agricultural Science and Technology Innovation Project of Hunan Province (2023CX63, 2024CX108), Earmarked fund for HARS- Chinese medicinal materials (HARS-11).

4. In this instance it seems there may be acceptable restrictions in place that prevent the public sharing of your minimal data. However, in line with our goal of ensuring long-term data availability to all interested researchers, PLOS’ Data Policy states that authors cannot be the sole named individuals responsible for ensuring data access (http://journals.plos.org/plosone/s/data-availability#loc-acceptable-data-sharing-methods).

Reviewers' comments:

Reviewer's Responses to Questions

**Comments to the Author**

1. Is the manuscript technically sound, and do the data support the conclusions?

Reviewer #1: Yes

Reviewer #2: Yes

2. Has the statistical analysis been performed appropriately and rigorously?

Reviewer #1: Yes

Reviewer #2: I Don't Know

3. Have the authors made all data underlying the findings in their manuscript fully available?

Reviewer #1: Yes

Reviewer #2: No

4. Is the manuscript presented in an intelligible fashion and written in standard English?

Reviewer #1: Yes

Reviewer #2: Yes

Reviewer #1: The aim of this study was to investigate the mechanism of color and flavor formation of Polygonatum cyrtonema Hua (PCH) during treatment, and the browning degree and major components, including polysaccharides, monosaccharides, 5-HMF, and CML, were analyzed for PCH. Meanwhile, LC-MS and HS-SPME-GCMS were utilized to reveal the changes of the components at different stages of treatment, and their correlations were evaluated. Some interesting results were obtained by comparing the differences in the composition of the constituents at different stages of steaming and drying treatments, and these results have certain reference value for revealing the formation mechanism of color and flavor during the processing of PCH. However, there are still some problems in the manuscript that need to be explained more clearly or improved.

1. Highlights section: Line 14: “The processing of PCH does not significantly reduce pharmacodynamics.” This sentence needs to be revised because there is no mention of pharmacodynamic evaluation in the manuscript.

2. Line 15~23: The abstract needs to be rewritten to be more specific about the results of the study.

3. Abbreviations that appear for the first time should include an explanation of the full name.

4. Introduction section: Line 52~54: “However, this hypothesis has some limitations. Firstly, the Maillard reaction uses monosaccharides and amino acids as substrate, and caramelization reactions could occur under the same conditions. The substrates and products of the two reactions are different.” Please clearly explain whether the substrates of the two reactions are the same or not. Line 63~65: “This implies that the color and flavor formation of processed PCHs is surely based on the consumption of water-soluble polysaccharides, and the processing will inevitably lead to a significant reduction in the pharmacological activities of PCHs.” This sentence needs additional cited references.

5. Line 119: In the antioxidant activity test, the concentration of the extract sample needs to be supplemented.

6. Line 197~198: “There was a significant reduction in relative lightness after each drying stage, with a slight recovery following steaming. (Fig. 2B).” Is it possible to try to explain the reasons for this pattern of change?

7. In order to better explain the mechanism of color and flavor formation of PCH, it is personally recommended to give the analytical results of LC-MS and GC-MS, combined with the representative end-products of caramelization reaction and Maillard reaction, and then further compare and analyze them, so that the results obtained will be more convincing.

8. The effect of the treatment of PCH on the quality (including color and flavor) has also been able to be reported in the previous literature, and the results of the previous reports should be appropriately cited or compared in this paper, to find out the trends of the same or different contents of the characteristic components, which will also provide more evidence for this paper on the formation of color and flavor proposed mechanisms.

Besides

“Supplementary Material 1” should be revised as “Supplementary Table 1”, and in the table, “yield (%)” is representing the amount of which substance? Please express it clearly.

Abbreviations for Polygonatum cyrtonema Hua need to be consistent in manuscripts, PCH or PCHs?

G0 and GO are used confusingly for the description of standard groups and should be standardized as “G0”.

Line 141: sett?

Line 298: (Fig. 8F and 8G)?

Line 331: Table 2, 5-HMFA? 5-MFA?

Line 335: “LC-MS” should be revised to “HS-SPME-GCMS”

In the Fig. 2, Note: C “The browing degree……” should be revised as “The browning degree……”

The “note” content should be included with the figure or table and not presented separately in the text.

Reviewer #2: This manuscript addresses a relevant and timely topic: the biochemical changes that occur during the processing of Polygonatum cyrtonema Hua (PCH), a plant widely consumed as both food and traditional medicine in China. Specifically, the study explores whether the observed shifts in taste and color, transitioning from bitter and yellow to sweet and brown, are primarily due to Maillard reactions (MR) or caramelization.

While the research question is of interest, the manuscript in its current form contains several weaknesses that preclude publication at this stage.

Introduction

- The authors aim to challenge the previously proposed assumption that Maillard reactions are responsible for the changes in color and flavor during PCH processing. However, the argument is not clearly structured or convincingly presented.

- The claim that MR and its products (MRPs) pose a food safety risk is not substantiated. MRPs are present in many everyday foods without posing health risks under normal conditions. Therefore, the rationale for questioning the MR mechanism is weak.

- The practical relevance of distinguishing between MR and caramelization is not sufficiently articulated. Beyond basic scientific interest, are there implications for food safety, quality control, or traditional processing validation?

Results

- Reported changes in protein, amino acid, flavonoid, and carbohydrate content are presented only as percentages. Without absolute concentrations (at least for starting values), it is impossible to assess molar ratios or interpret the data quantitatively.

- In line 335: GC-MS?

- The labeling in Figure 4 is too small and not legible.

Discussion

- The conclusion that MR is not primarily responsible for color and aroma formation, and that potentially harmful MRPs can thus be neglected, is overly assertive. Since specific (toxic) MRPs were not identified or quantified, this statement remains speculative.

Data Availability

- The authors state that some omics data are part of ongoing projects and cannot be made public. The authors say that they provide access to confidential data for researchers who meet the criteria. Which criteria? A transparent data sharing policy should define the access criteria and, at minimum, provide metadata or representative datasets to ensure reproducibility.

The manuscript addresses a valuable research question, but it requires substantial revisions to meet the standards of scientific rigor.

**Do you want your identity to be public for this peer review?** For information about this choice, including consent withdrawal, please see our Privacy Policy

Reviewer #1: No

Reviewer #2: No

---

## [Author Response · Author response to Decision Letter 1]

17 Sep 2025

Dear Editor & Reviewers:

Thank you sincerely for your patient comment and suggestions about our manuscript “Study on the Formation Mechanism of Color and Flavor during the Processing of Polygonatum cyrtonema Hua (PCH)” (PONE-D-25-34418). These comment and suggestions are all valuable and helpful for revising and improving our manuscript, and we have carefully revised the manuscript according to these comment, The red font represents the modified content in manuscript.The following is the Response List of comment.

Reviewer #1:

Comment 1. Highlights section: Line 14: “The processing of PCH does not significantly reduce pharmacodynamics.” This sentence needs to be revised because there is no mention of pharmacodynamic evaluation in the manuscript.

Response: In this study, efficacy evaluation was rigorously conducted according to the experimental design framework, encompassing multiple aspects including in vitro antioxidant properties, anti-aging activity, and anti-inflammatory effects. Under the original research structure, we initially planned to compile anti-aging activity data alongside other efficacy evaluations for separate publication, as we hypothesized that PCH anti-aging effects might stem from its specific anti-inflammatory manifestations. However, through comprehensive analysis, we found that the correlation between PCH anti-aging and anti-inflammatory activities was insufficient to support this hypothesis. Consequently, these data are now considered appropriate for discussion and application within this study.

The anti-aging efficacy of Polygonatum cyrtonema Hua stands as one of its primary pharmacological properties. In the revised manuscript, we have incorporated and utilized relevant research data from this study. We treated HaCaT cells with UVB radiation and assessed senescence progression using SA-β-Gal staining. Experimental results demonstrated that within five processing cycles, there was no statistically significant difference in anti-aging activity between PCH samples (p values > 0.01).

Comment 2. Line 15~23: The abstract needs to be rewritten to be more specific about the results of the study.

Response: Thank you for your comment. We agree that the current abstract does not adequately represent our research, so the abstract has been rewritten to ensure that it better reflects the research content.

Comment 3. Abbreviations that appear for the first time should include an explanation of the full name.

Response: Thank you for your careful examination of the manuscript. We found that some abbreviations were missing from the full name when they first appeared, and this problem has been corrected in the revised manuscript.

Comment 4. Introduction section: Line 52~54: “However, this hypothesis has some limitations. Firstly, the Maillard reaction uses monosaccharides and amino acids as substrate, and caramelization reactions could occur under the same conditions. The substrates and products of the two reactions are different.” Please clearly explain whether the substrates of the two reactions are the same or not. Line 63~65: “This implies that the color and flavor formation of processed PCHs is surely based on the consumption of water-soluble polysaccharides, and the processing will inevitably lead to a significant reduction in the pharmacological activities of PCHs.” This sentence needs additional cited references.

Response: We appreciate your feedback on this section. We have recognized that the wording in our previous manuscript inadequately conveyed our intended meaning. While Maillard reactions and caramelization are sometimes confused, their fundamental mechanisms differ significantly: Maillard reactions involve the condensation of carbonyl compounds with amino compounds, whereas caramelization occurs without amino involvement – a key distinction we have identified. The nitrogen-containing compounds in our products likely originate from Strecker degradation of α-amino and α-diketo compounds in Maillard reactions, including pyrazine, acrylamide, and CML. In contrast, oxygen-containing compounds such as furfural and maltol are directly produced through caramelization without direct connection to Maillard reactions. We have revised the relevant sections of the manuscript (marked in red) and supplemented them with necessary references.

Comment 5. Line 119: In the antioxidant activity test, the concentration of the extract sample needs to be supplemented.

Response: We appreciate your feedback regarding the inadequacies in this section. The experimental samples were prepared by re-dissolving dried extracts, with all concentrations set at 0.05 g/mL. However, our manuscript failed to properly describe this methodology or specify the concentration values. We have revised the relevant sections of the manuscript, with the updated content highlighted in red for clarity.

Comment 6. Line 197~198: “There was a significant reduction in relative lightness after each drying stage, with a slight recovery following steaming. (Fig. 2B).” Is it possible to try to explain the reasons for this pattern of change?

Response: Thank you for bringing this critical issue to our attention, which we have indeed considered during the article's development. Our research hypothesis suggests that the observed increase in lightness after steaming may be related to the loss of 5-HMF from the sample surface through water vapor evaporation. The browning degree data indirectly supports this hypothesis. Although we initially planned to conduct a cross-sectional brightness assessment to validate this conclusion, repeated attempts to obtain viable sample sections proved unsuccessful due to the fragility of dried specimens. Consequently, the original manuscript did not address the underlying causes of this phenomenon. In this revision, we have incorporated your suggestions by referencing literature on similar phenomena to explain this aspect. The revised sections are highlighted in red for clarity.

Comment 7. In order to better explain the mechanism of color and flavor formation of PCH, it is personally recommended to give the analytical results of LC-MS and GC-MS, combined with the representative end-products of caramelization reaction and Maillard reaction, and then further compare and analyze them, so that the results obtained will be more convincing.

Response: Thank you for your feedback. In our earlier manuscript, the data from LC-MS and HS-SPME-GCMS were not fully utilized. Through this revision, we identified four representative compounds based on differences in Maillard reaction and caramelization products. The results showed that the content changes of pyrazine and acrylamide—representing Maillard reactions—were relatively minor compared to those of furfural and furan, which signify caramelization processes. Moreover, these compounds exhibited significant differences in safety profiles, a finding that further supports our conclusions.

Comment 8. The effect of the treatment of PCH on the quality (including color and flavor) has also been able to be reported in the previous literature, and the results of the previous reports should be appropriately cited or compared in this paper, to find out the trends of the same or different contents of the characteristic components, which will also provide more evidence for this paper on the formation of color and flavor proposed mechanisms.

Response: Thank you for your comment. We have compared the change laws of polysaccharides, monosaccharides, amino acids, proteins, 5-HMF, and other compounds with those reported in existing literature, and have added appropriate references. The revised sections are marked in red in the manuscript.

Other comment: 1) Supplementary Material 1” should be revised as “Supplementary Table 1”, and in the table, “yield (%)” is representing the amount of which substance? Please express it clearly. 2) Abbreviations for Polygonatum cyrtonema Hua need to be consistent in manuscripts, PCH or PCHs?3) G0 and GO are used confusingly for the description of standard groups and should be standardized as “G0”.4) Line 141: sett?; Line 298: (Fig. 8F and 8G)?; Line 331: Table 2, 5-HMFA? 5-MFA?; Line 335: “LC-MS” should be revised to “HS-SPME-GCMS”; In the Fig. 2, Note: C “The browing degree……” should be revised as “The browning degree……”; The “note” content should be included with the figure or table and not presented separately in the text.

Response: Thank you for your comment. We have corrected the above errors and marked them in red in the manuscript.

Reviewer #2:

Comment 1. Introduction: The authors aim to challenge the previously proposed assumption that Maillard reactions are responsible for the changes in color and flavor during PCH processing. However, the argument is not clearly structured or convincingly presented. The claim that MR and its products (MRPs) pose a food safety risk is not substantiated. MRPs are present in many everyday foods without posing health risks under normal conditions. Therefore, the rationale for questioning the MR mechanism is weak. The practical relevance of distinguishing between MR and caramelization is not sufficiently articulated. Beyond basic scientific interest, are there implications for food safety, quality control, or traditional processing validation?

Response: Thank you for your Comment. We have made the following revisions: First, we expanded the description of caramelization and Maillard reaction product types in the Introduction section. Second, we conducted a comprehensive analysis of LC-MS and HS-SPME-GCMS data to identify four representative new compounds that can represent both reactions, summarizing their variation patterns. Key findings include: 1) The content of acrylamide and other compounds with reported safety risks showed relatively minor changes; 2) Compounds representing caramelization, such as furfural and furan, experienced significant variations, yet these are deemed safe by WHO's technical report ("Evaluation of Certain Food Additives and Contaminants"); 3) Since caramelization dominates the processing of polyhydroxychlorides (PCH), similar processing methods pose no safety risks.

Comment 2. Results: Reported changes in protein, amino acid, flavonoid, and carbohydrate content are presented only as percentages. Without absolute concentrations (at least for starting values), it is impossible to assess molar ratios or interpret the data quantitatively. In line 335: GC-MS? The labeling in Figure 4 is too small and not legible.

Response: Thank you for your comment. We recognize that there are indeed problems with these parts of the diagram, so we have modified them according to your suggestions.

Comment 3. Discussion: The conclusion that MR is not primarily responsible for color and aroma formation, and that potentially harmful MRPs can thus be neglected, is overly assertive. Since specific (toxic) MRPs were not identified or quantified, this statement remains speculative.

Response: Thank you again for your feedback. In the earlier manuscript, this conclusion was indeed unreliable. We also acknowledge that flavor and safety risks are not directly related. In the revised version, we have supplemented this conclusion through methods such as relative quantification of key representative compounds and inclusion of bioactivity evaluation experiments.

Comment 4. Data Availability: The authors state that some omics data are part of ongoing projects and cannot be made public. The authors say that they provide access to confidential data for researchers who meet the criteria. Which criteria? A transparent data sharing policy should define the access criteria and, at minimum, provide metadata or representative datasets to ensure reproducibility.

Response: Thank you for your comment. We have recognized the importance of ensuring data accessibility and resolved the original limitation of data disclosure. All data involved in this manuscript has now been uploaded to Figshare (doi: 10.6084/m9.figshare.30039019) and is publicly accessible.

Journal Requirements:

Comment 1. Please ensure that your manuscript meets PLOS ONE's style requirements, including those for file naming.

Response: This manuscript has been revised in accordance with the requirements of the author's instructions on content, format and attachments.

Comment 2. PLOS requires an ORCID iD for the corresponding author in Editorial Manager on papers submitted after December 6th, 2016. Please ensure that you have an ORCID iD and that it is validated in Editorial Manager. To do this, go to ‘Update my Information’ (in the upper left-hand corner of the main menu), and click on the Fetch/Validate link next to the ORCID field. This will take you to the ORCID site and allow you to create a new iD or authenticate a pre-existing iD in Editorial Manager.

Response: The ORCID iD of the corresponding author has been submitted and validated.

Comment 3. Thank you for stating the following financial disclosure:This research was funded by the Agricultural Science and Technology Innovation Project of Hunan Province (2023CX63, 2024CX108), Earmarked fund for HARS- Chinese medicinal materials (HARS-11).Please state what role the funders took in the study. If the funders had no role, please state: "The funders had no role in study design, data collection and analysis, decision to publish, or preparation of the manuscript."If this statement is not correct you must amend it as needed.Please include this amended Role of Funder statement in your cover letter; we will change the online submission form on your behalf.

Response: This research was supported by the Hunan Academy of Agricultural Sciences. The funding party did not participate in the design of the study, data collection and analysis, publication decisions, or manuscript preparation. We have included relevant explanations in the manuscript.

Comment 4. In this instance it seems there may be acceptable restrictions in place that prevent the public sharing of your minimal data. However, in line with our goal of ensuring long-term data availability to all interested researchers, PLOS’ Data Policy states that authors cannot be the sole named individuals responsible for ensuring data access

Response: The date involved in the manuscript have published and uploaded it to Figshare (doi: 10.6084/m9.figshare.30039019), or contact the data custodian to obtain the data by sending an email (tangym1208@163.com). All the data can be obtained by our own through links now. When the manuscript was uploaded before, the data set was not disclosed because it involved the research of many projects of our team. Now this problem has been solved. All data owners to reach an agreement.

Comment 5. When completing the data availability statement of the submission form, you indicated that you will make your data available on acceptance. We strongly recommend all authors decide on a data sharing plan before acceptance, as the process can be lengthy and hold up publication timelines. Please note that, though access restrictions are acceptable now, your entire data will need to be made freely accessible if your manuscript is accepted for publication. This policy applies to all data except where public deposition would breach compliance with the protocol approved by your research ethics board. If you are unable to adhere to our open data policy, please kindly revise your statement to explain your reasoning and we will seek the editor's input on an exemption. Please be assured that, once you have provided your new statement, the assessment of your exemption will not hold up the peer review process.

Response: The date involved in the manuscript have published. Therefore, the data availability statement in the submission system is modified accordingly.

Comment 6. Please include captions for your Supporting Information files at the end of your manuscript, and update any in-text citations to match accordingly.

Response: The Supporting Information has been added at the end of the manuscript as required by the guidelines in the Supporting Information.

Comment 7. If the reviewer comment include a recommendation to cite specific previously published works, please review and evaluate these publications to determine whether they are relevant and should be cited. There is no requirement

---

## [Decision Letter · Decision Letter 1]

10 Nov 2025

Dear Dr. Song,

Thank you for submitting your manuscript to PLOS ONE. After careful consideration, we feel that it has merit but does not fully meet PLOS ONE’s publication criteria as it currently stands. Therefore, we invite you to submit a revised version of the manuscript that addresses the points raised during the review process.

We look forward to receiving your revised manuscript.

Kind regards,

José M. Alvarez-Suarez

Academic Editor

PLOS ONE

Journal Requirements:

Reviewers' comments:

Reviewer's Responses to Questions

**Comments to the Author**

Reviewer #3: All comments have been addressed

Reviewer #4: (No Response)

2. Is the manuscript technically sound, and do the data support the conclusions?

Reviewer #3: Yes

Reviewer #4: Yes

3. Has the statistical analysis been performed appropriately and rigorously?

Reviewer #3: Yes

Reviewer #4: Yes

4. Have the authors made all data underlying the findings in their manuscript fully available?

Reviewer #3: Yes

Reviewer #4: Yes

5. Is the manuscript presented in an intelligible fashion and written in standard English?

Reviewer #3: Yes

Reviewer #4: Yes

Reviewer #3: A thorough review is required. The study is interesting and publishable, but key issues need to be addressed: the statistical reporting, the lack of controls to support claims regarding the mechanism, the reliability of metabolite identification, and data availability before supporting claims (especially that caramel—not Millard—is the primary cause of tanning, and that PCH-treated products are safe).

Insufficient details regarding replication and biological/analytical statistics. The methods do not specify the number of biological replicates per set for most tests (e.g., LC-MS, GC-MS, 5-HMF, CML, antioxidant assays).

Statistical approach: The authors indicate the use of SPSS and LSD (lowest significant difference), but they do not describe the correction of multiple tests for thousands of traits in LC-MS/GC-MS (using log2FC >|1| and p<0.01). Which test was applied? FDR/Benjamini-Hochberg?

The high correlation between tanning degree and 5-HMF (reported r = 0.96) suggests that caramelization is the dominant chemical pathway for color formation, but it is not conclusive proof. This correlation may reflect co-production, co-depletion of the substrate, or simply phase effects. Mechanistic claims (caramelization > Millard for color) require experimental evidence (model systems or kinetic data), not just correlations.

5-HMF is used as a marker for caramelization; CML is used for Maillard. Both markers can be produced via multiple pathways (e.g., 5-HMF can arise under certain Maillard pathways; CML measurement using ELISA may suffer from cross-reactivity). This manuscript does not discuss these caveats or validate the markers in this specific matrix.

Evaporation at 105 °C (1 hour) and drying at 55 °C (6 hours) were used (Methods). Typical caramelization temperatures are higher; the authors cite studies suggesting that caramelization can occur at lower temperatures. However, to claim that drying induces caramelization, direct evidence must be provided that the thermal history (time × temperature), and not the effect of water activity or concentration, for example, is what induces the accumulation of 5-HMF. A range of controlled model reactions (pure sugar ± amino acid at that temperature and humidity) or measurements of the sample's internal temperature and water activity support this claim.

The authors state that the processed PCH compounds are safe because the measured 5-HMF concentration is low (5.13 ± 0.39 mg/L), and they cite a toxicology paper that sets the doses for adverse effects between 80 and 100 mg/kg. However, estimating human exposure requires consumption data, concentration units (mg/g dry weight), and comparison to regulatory or permissible values (not just acute toxicity in mg/kg). A safety statement is not currently supported. Please provide the concentration per serving, intake estimates, and citations from appropriate risk assessments or regulatory limits.

The data availability statement indicates that some omics data cannot be shared publicly due to other ongoing projects. The authors should either deposit the spectral data (raw LC-MS, GC-MS) in public repositories (MetaboLights, GNPS, MassIVE, etc.) or provide a strong justification for restricting access and a clear access mechanism.

Criteria for Metabolite Identification and Reporting: For LC-MS/GC-MS identifications, the manuscript provides numbers for the detected properties and some named compounds, but it does not precisely define the confidence levels of the identification (e.g., MSI level 1 to 4), nor does it provide examples of spectra, retention periods, or reference standards for key markers (5-HMF, pyrazines, and furanones). For metabolites used to support mechanistic claims, an MSI level 1 identification (the original standard) or an MS/MS spectral match with the library and retention index must be provided.

Reviewer #4: The authors have adequately addressed all comments raised in a previous round of review, i think it can be acceptable for publish in PLUS-ONE after minor revision.

line 32 what is the Maillardization?

Why are Figures 1 and 2 placed at the end of the manuscript?

Please provide a comprehensive summary or conclusion after the section of discussion

**Do you want your identity to be public for this peer review?** For information about this choice, including consent withdrawal, please see our Privacy Policy

Reviewer #3: No

Reviewer #4: No

---

## [Author Response · Author response to Decision Letter 2]

24 Dec 2025

Dear Editor & Reviewers:

Thank you sincerely for your patient comment and suggestions about our manuscript “Study on the Formation Mechanism of Color and Flavor during the Processing of Polygonatum cyrtonema Hua (PCH)” (PONE-D-25-34418). These comment and suggestions are all valuable and helpful for revising and improving our manuscript, and we have carefully revised the manuscript according to these comment, The red font represents the modified content in manuscript.The following is the Response List of comment.

Reviewer #1:

Comment 1. The methods do not specify the number of biological replicates per set for most tests (e.g., LC-MS, GC-MS, 5-HMF, CML, antioxidant assays).

Response: The experiments included LC-MS, GC-MS, 5-HMF, CML, and antioxidant assays, as previously submitted experimental data to the online platform (doi: 10.6084/m9.figshare.30039019), with all replicates performed three times. The actual number of replicates for each experiment has been noted in the manuscript according to the experimental conditions, and corresponding modifications are highlighted in red.

Comment 2. The authors indicate the use of SPSS and LSD (lowest significant difference), but they do not describe the correction of multiple tests for thousands of traits in LC-MS/GC-MS (using log2FC >|1| and p<0.01). Which test was applied? FDR/Benjamini-Hochberg?

Response: In the LC-MS and CG-MS analyses, the original results were based on the Variable Importance in Projection (VIP) derived from the OPLS-DA model. The fold change (FC) from univariate analysis was further combined to screen for differential metabolites. The criteria for differential metabolites were log2FC> |1| and VIP> 1. The corresponding explanations are provided in the 'Statistical analysis' section of the manuscript, highlighted in red.

Comment 3. The high correlation between tanning degree and 5-HMF (reported r = 0.96) suggests that caramelization is the dominant chemical pathway for color formation, but it is not conclusive proof. This correlation may reflect co-production, co-depletion of the substrate, or simply phase effects. Mechanistic claims (caramelization > Millard for color) require experimental evidence (model systems or kinetic data), not just correlations. 5-HMF is used as a marker for caramelization; CML is used for Maillard. Both markers can be produced via multiple pathways (e.g., 5-HMF can arise under certain Maillard pathways; CML measurement using ELISA may suffer from cross-reactivity). This manuscript does not discuss these caveats or validate the markers in this specific matrix. Evaporation at 105℃(1 hour) and drying at 55℃ (6 hours) were used (Methods). Typical caramelization temperatures are higher; the authors cite studies suggesting that caramelization can occur at lower temperatures. However, to claim that drying induces caramelization, direct evidence must be provided that the thermal history (time × temperature), and not the effect of water activity or concentration, for example, is what induces the accumulation of 5-HMF. A range of controlled model reactions (pure sugar ± amino acid at that temperature and humidity) or measurements of the sample's internal temperature and water activity support this claim.

Response: We also consider this a critical issue. The ability of caramelization reactions without amino compounds to occur at lower temperatures and independently produce 5-HMF that provide color for PCHs directly determines the reliability of the conclusion. Therefore, we isolated polysaccharide components from the experimental samples, removed proteins through five rounds of Sevag method until the samples showed no absorption peaks at 280 nm, and then processed them similarly to PCH samples. The brown solid obtained after processing was volumetrically standardized to 10 mL, at which point the 5-HMF concentration was 10.80 ± 0.31 mg/L, and the color was similar to that of the processed PCH. Additionally, a control group was set by adding 7% lysine (close to the ratio of polysaccharides to amino acids in PCH raw materials) to Polygonatum polysaccharides. After processing, the 5-HMF concentration was 12.14 ± 0.45 mg/L. The results indicate that caramelization reactions can occur under these conditions and serve as the primary factor for PCH coloring. This result has been added to the manuscript and marked in red, while the original data has been uploaded to the online website.

Comment 4. The authors state that the processed PCH compounds are safe because the measured 5-HMF concentration is low (5.13 ± 0.39 mg/L), and they cite a toxicology paper that sets the doses for adverse effects between 80 and 100 mg/kg. However, estimating human exposure requires consumption data, concentration units (mg/g dry weight), and comparison to regulatory or permissible values (not just acute toxicity in mg/kg). A safety statement is not currently supported. Please provide the concentration per serving, intake estimates, and citations from appropriate risk assessments or regulatory limits.

Response: This is a important comment. We recognize that although the current data supports the evaluation of the safety threshold of PCH and 5-HMF, the excessively ambiguous wording hindered readers' understanding. Consequently, we have supplemented the relevant descriptions. Based on the existing data in the manuscript, the 5-HMF content in PCH samples does not exceed 5.13 mg per 100 g. According to the safety threshold provided in the references, the daily safe intake for a adult (60 kg) should not exceed 6 g. In traditional PCH consumption, the recommended daily intake as a herbal medicine is 10 g, resulting in 0.51 mg of 5-HMF. For regular food consumption, the intake generally does not exceed 500 g, yielding 0.026 g of 5-HMF. Both values are three orders of magnitude lower than the threshold at which adverse reactions are likely to occur. Thus, it can be concluded that 5-HMF in PCH does not present a safety risk. This revised section has been incorporated into the updated manuscript and marked in red.

Comment 5. The data availability statement indicates that some omics data cannot be shared publicly due to other ongoing projects. The authors should either deposit the spectral data (raw LC-MS, GC-MS) in public repositories (MetaboLights, GNPS, MassIVE, etc.) or provide a strong justification for restricting access and a clear access mechanism.

Response: Following the previous revision, all experimental data referenced in the manuscript are now publicly available on the online website (doi: 10.6084/m9.figshare.30039019). Additionally, the supplementary experimental results on 5-HMF added in this revision have been uploaded to FigShare.

Comment 6. Criteria for Metabolite Identification and Reporting: For LC-MS/GC-MS identifications, the manuscript provides numbers for the detected properties and some named compounds, but it does not precisely define the confidence levels of the identification (e.g., MSI level 1 to 4), nor does it provide examples of spectra, retention periods, or reference standards for key markers (5-HMF, pyrazines, and furanones). For metabolites used to support mechanistic claims, an MSI level 1 identification (the original standard) or an MS/MS spectral match with the library and retention index must be provided.

Response: The MS spectrometry results in the manuscript were processed by comparing them with our proprietary database based on the ms-dial dataset (ESI (+) -MS/MS from authentic standards) and calibrated using the HMDB and PubChem databases. Information such as mass-to-charge ratios, retention times, molecular weights, ion types, and comparison scores has been included in the publicly available data (014-raw date of LC-MS and 013-raw date of HS-SPME-GCMS), which can be downloaded via the FigShare website. Additional information about the datasets used has been added to the manuscript and highlighted in red.

Reviewer #2:

Comment 1. The authors have adequately addressed all comments raised in a previous round of review, i think it can be acceptable for publish in PLUS-ONE after minor revision. line 32 what is the Maillardization?

Response: This is a typographical error. The correct term is "Maillard reactions".

Comment 2. Why are Figures 1 and 2 placed at the end of the manuscript?

Response: This is the automatically adjusted position in the submission system. No figure was added to the original manuscript file, and the figures were uploaded separately.

Comment 3. Please provide a comprehensive summary or conclusion after the section of discussion.

Response: We recognize that the lack of a discussion section for summarization substantially diminished the readability of the article. Consequently, we have incorporated your suggestion to include a "conclusion" section at the end of the manuscript. We are grateful for your comment.

Journal Requirements:

Comment 1. Please review your reference list to ensure that it is complete and correct. If you have cited papers that have been retracted, please include the rationale for doing so in the manuscript text, or remove these references and replace them with relevant current references. Any changes to the reference list should be mentioned in the rebuttal letter that accompanies your revised manuscript. If you need to cite a retracted article, indicate the article’s retracted status in the References list and also include a citation and full reference for the retraction notice.

Response: We have verified the references cited in the manuscript to ensure that no withdrawn literature is included. Additionally, based on the reviewers' comments, we supplemented the experiments and added two references to corroborate the obtained data.

We appreciate your time and patience for editing our manuscript and giving invaluable suggestions to improve quality of the manuscript.

Yours sincerely,

Corresponding author:

Rong Song

E-mail: songrong0205@163.com

---

## [Decision Letter · Decision Letter 2]

15 Feb 2026

Study on the Formation Mechanism of Color and Flavor during the Processing of Polygonatum cyrtonema Hua (PCH)

PONE-D-25-34418R2

Dear Dr. Song,

We’re pleased to inform you that your manuscript has been judged scientifically suitable for publication and will be formally accepted for publication once it meets all outstanding technical requirements.

Kind regards,

José M. Alvarez-Suarez

Academic Editor

PLOS One

Additional Editor Comments (optional):

Reviewers' comments:

Reviewer's Responses to Questions

**Comments to the Author**

Reviewer #4: All comments have been addressed

2. Is the manuscript technically sound, and do the data support the conclusions?

Reviewer #4: Yes

3. Has the statistical analysis been performed appropriately and rigorously?

Reviewer #4: Yes

4. Have the authors made all data underlying the findings in their manuscript fully available?

Reviewer #4: Yes

5. Is the manuscript presented in an intelligible fashion and written in standard English?

Reviewer #4: Yes

Reviewer #4: (No Response)

**Do you want your identity to be public for this peer review?** For information about this choice, including consent withdrawal, please see our Privacy Policy

Reviewer #4: No

---

## [Editor Report · Acceptance letter]

PONE-D-25-34418R2

PLOS One

Dear Dr. Song,

I'm pleased to inform you that your manuscript has been deemed suitable for publication in PLOS One. Congratulations! Your manuscript is now being handed over to our production team.

Kind regards,

on behalf of

Professor José M. Alvarez-Suarez

Academic Editor

PLOS One